# Isolation of quartz for cosmogenic in situ $^{14}$C analysis

Keir A. Nichols[1], Brent M. Goehring[1]

[1]Department of Earth and Environmental Sciences, Tulane University, New Orleans, LA, 70118, USA

*Correspondence to*: Keir A. Nichols (knichol3@tulane.edu)

**Abstract** Froth flotation is a commonly used procedure for separating feldspars and micas from quartz for the preparation of quartz mineral separates to carry out cosmogenic nuclide analysis. Whilst extracting carbon from quartz we observed in situ carbon-14 ($^{14}$C) concentrations which were anomalously high and in excess of theoretical geological maximum concentrations. Further etching of sample material reduced carbon yields and $^{14}$C concentrations, yet the latter remained unrealistically high. When quartz from the original whole rock sample was isolated in our laboratory, we observed even lower carbon yields and

geologically plausible in situ $^{14}$C concentrations. After ruling out unlikely geological scenarios and systematic measurement issues, we decided to investigate the quartz isolation procedure as a potential source of $^{14}$C contamination. We hypothesised that laurylamine (dodecylamine), an organic compound used as part of the froth flotation procedure, elevates $^{14}$C concentrations if residual laurylamine is present. We demonstrate that laurylamine has a $^{14}$C modern carbon source and thus has the potential to influence in situ $^{14}$C measurements if present in minute but measurable quantities. Furthermore, we show

that insufficient sample etching results in contaminant $^{14}$C persisting through step heating of quartz that is subsequently collected with the in situ component released at 1100 °C. We demonstrate that froth flotation contaminates in situ $^{14}$C measurements. We provide guidelines for the preparation of quartz based on methods developed in our laboratory and demonstrate that all froth flotation-derived carbon and $^{14}$C is removed when applied. We recommend that the procedures presented be used at a minimum when using froth flotation to isolate quartz for in situ $^{14}$C measurements.

**1.0 Introduction**

      In the course of extracting carbon from quartz we have, on multiple occasions, observed concentrations of in situ $^{14}$C that were anomalously high and in excess of geologically plausible maximum concentrations. We hypothesise that the elevated in situ $^{14}$C concentrations are sourced from part of the widely used mineral separation procedure known as froth flotation, a process that relies on three organic compounds; laurylamine (also known as dodecylamine, $C_{12}H_{27}N$), eucalyptol ($C_{10}H_{18}O$),

and acetic acid ($C_2H_4O_2$). Our observations, combined with a desire to continue use of froth flotation for the benefits it provides during quartz separation, form the motivation for this paper. In this study we explore both the potential influence that quartz isolation procedures have on resulting measured $^{14}$C concentrations as well as procedures to minimise potential contamination during use of froth flotation.

Froth flotation is a method by which feldspars and, to a lesser extent, micas are separated from quartz (Herber, 1969). The method precedes acid etching as part of the quartz isolation process for cosmogenic nuclide analysis and is used by numerous cosmogenic nuclide laboratories. It is useful for samples containing large proportions of feldspars and vastly reduces the resources required to etch samples. A motivating factor for this study was the realisation that froth flotation introduces carbon to sample material through the use of three aforementioned organic compounds. Use of the organic compounds was previously of no consequence as the method was primarily used to isolate quartz for the measurement of $^{10}$Be and $^{26}$Al. There is no standard procedure for froth flotation or post-froth flotation sample etching. As a result, different laboratories use various quantities of laurylamine, eucalyptol, and acetic acid, as well as varying etching procedures, which complicates the matter further. The carbon content, and especially the $^{14}$C content, of the three organic compounds used in our laboratory has yet to be measured, thus the potential for contamination of in situ $^{14}$C measurements is unquantified.

In the first part of this work, we summarise the froth flotation procedure as well as the overall quartz isolation process used for in situ $^{14}$C analysis. We describe the range of methodologies used today, and detail those used at Tulane. We then describe the initial measurements that led us to hypothesise that froth flotation could be causing contamination of in situ $^{14}$C results. Finally, we describe the methodology and results of a systematic study that demonstrates laurylamine contains modern carbon, that froth flotation does contaminate samples with regards to both $^{14}$C and carbon in general, and that contaminant $^{14}$C can be removed with sufficient sample etching. We demonstrate that the post-froth flotation etching methodology used in our laboratory ensures that quartz is isolated effectively and without influencing the resulting in situ $^{14}$C measurements. We conclude that froth flotation should be applied with care if in situ $^{14}$C is to be measured, and that the post-froth flotation etching methodology described below should be applied at a minimum to ensure that samples are free of contaminant $^{14}$C from froth flotation.

## 1.1 Froth flotation and the isolation of quartz from whole rock material

### 1.1.1 Pre-Froth Flotation

Prior to froth flotation, whole rock material is typically crushed, milled and sieved to isolate the 250 - 500 μm size fraction. This is then rinsed with tap or deionised water to remove any fine grain-sized material. At this point samples are ready for froth flotation, although we commonly first dry samples so that a magnetic mineral separation can be performed to remove any mafic material present prior to frothing, which we find improves overall frothing efficiency. The sample is ready for froth flotation following the removal of fine grain-sized material and the optional magnetic separation.

Our method for froth flotation is largely based on that used at PRIME Lab (http://www.physics.purdue.edu/primelab/MSL/froth_floatation.html). The first stage of froth flotation is the conditioning of sample material with dilute (< 5% v/v) hydrofluoric acid (HF). Conditioning the sample makes the feldspar (and mica) grains hydrophobic and the quartz grains hydrophilic, which is key to the separation process. We condition each sample in a 1 L

Nalgene bottle with enough 5 % HF/HNO$_3$ to saturate and cover the sample, without agitation beyond gently swirling the bottle a few times. The sample is left to sit for no more than five minutes before decanting the acid solution and beginning froth flotation. Some laboratories condition the sample with dilute HF (1 to 5 %) for up to 60 minutes on a shaker table; we have found that five minutes with 5 % HF/HNO$_3$ is sufficient and improves separation efficiency.

### 1.1.2 Frothing Solution

Laurylamine is combined with glacial acetic acid by dissolution to form the frothing solution, typically at a 1:1 ratio. We combine approximately 300 ml of glacial acetic acid and 300 ml of laurylamine to form a 600 ml stock frothing solution. The frothing solution is then combined with water and carbonated or mixed with bubbly tap water. Some laboratories add the concentrated frothing solution directly to sample material, followed by the addition of carbonated or bubbly tap water. Other laboratories make the stock solution of glacial acetic acid and laurylamine and combine it with water before adding it to the sample material. For each sample, we combine approximately 6 ml of frothing solution with 20 L of water. The net concentration of both acetic acid and laurylamine in the frothing solution is 0.03% v/v. In terms of their purpose in the froth flotation procedure, laurylamine acts as a collector agent, or surfactant, and is thus required to separate the hydrophobic and hydrophilic mineral grains. Glacial acetic acid is used because laurylamine dissolves into it more readily compared to water, and it keeps the pH of the solution low.

### 1.1.3 Froth Flotation

A few drops of eucalyptus oil are added to the sample in a bowl (usually metal or plastic) before the frothing solution is applied to the sample. The eucalyptus oil holds the bubbles together to which the feldspar and mica grains attach. We use a hose connected to a soda-fountain carbonator to dispense the frothing solution. The now carbonated and dilute frothing solution is used to move the sample material from the 1 L bottle to the bowl. The frothing solution is then applied to the sample material in the bowl. The feldspar grains, owing to their hydrophobic nature, float to the top of the mixture whilst the quartz grains remain at the bottom. We apply 3 to 4 L of dilute frothing solution to the sample before waiting a few seconds and decanting the feldspar grains into a second bowl. The feldspar grains are usually discarded, though they may be saved for $^{36}$Cl analysis. The froth flotation procedure is repeated until most of the feldspar fraction has been removed or no additional separation of quartz and feldspar is accomplished. For a granitic sample of ca. 400 to 500 g, we find that five to six rounds of froth flotation are needed before either the froth flotation process is complete and the vast majority of feldspar has been removed, or froth flotation becomes less effective and the sample requires further conditioning. After reconditioning the sample in 5 % HF/HNO$_3$ for five minutes, additional rounds of froth flotation can be performed.

### 1.1.4 Post-Froth Flotation Acid Etching

Froth flotation is followed by etching the sample in HF or HF/HNO$_3$ to remove extraneous minerals, to partially dissolve or etch the quartz grains to remove meteoric cosmogenic nuclides, and to lower major ion concentrations (e.g., Fe, Ti, Al). Generally, the etching process follows a heavily modified version of the method of Kohl and Nishiizumi (1992). A typical procedure used by many laboratories involves first etching samples in 5 or 1 % HF or HF/HNO$_3$ on a shaker table or sample roller for multiple periods, followed by etching in 1 % HF or HF/HNO$_3$ in an ultrasonic bath. Some laboratories etch samples in an ultrasonic bath without the use of a sample roller or shaker table beforehand. Between etches, samples are rinsed with deionised water (i.e., 18.2 MΩ H$_2$O). The number of etches will vary due to factors such as sample lithology, amount of sample material, effectiveness of the froth flotation procedure, in addition to the varying minimum standard procedures for a given laboratory. For in situ $^{14}$C analysis, samples are often etched until they pass a visual test under a binocular microscope and the sample appears to be solely composed of quartz.

### 1.2 Initial Anomalous C-14 Measurements

Whilst measuring the in situ $^{14}$C concentration of glacial erratic samples as part of multiple projects we observed in situ $^{14}$C measurements that were in excess of geologically plausible maximum concentrations (Fig. 1). In each case the maximum concentration for a sample is set by the in situ $^{14}$C saturation concentration for the given sample location, shown in Fig. 1. The only way elevated in situ $^{14}$C concentrations could be explained is with an unlikely geomorphic scenario in which the samples were exposed at much higher elevations for a significant period of time before being rapidly transported to their sampling location. This phenomenon was described by Balco et al. (2016) and potentially observed by Balco et al. (2019). Subsequent elevated in situ $^{14}$C concentrations measured from bedrock samples led us to rule out this scenario as the sole source of the observed elevated in situ $^{14}$C concentrations, and we began to explore other explanations.

To investigate the cause or causes for the anomalously high in situ $^{14}$C measurements we performed additional measurements of in situ $^{14}$C concentrations of samples displaying elevated concentrations following additional etches in 1 % HF/HNO$_3$ for two 24-hour periods. Further etching resulted in unit yields comparable to the initial measurements and lower, but still anomalously high, $^{14}$C concentrations (Fig. 1) (Hillenbrand, unpub.). We note that the quartz from which the anomalously high in situ $^{14}$C concentrations and elevated carbon yields were measured was isolated at other laboratories that use slight variations in their quartz isolation procedures to ours. To investigate further, we measured the in situ $^{14}$C concentration from the same samples but isolated the quartz from whole rock material using our standard procedure (Sect. 1.1). With the exception of one sample, carbon yields were reduced (Fig. 2), and for all samples the resulting $^{14}$C concentrations were both lower and geologically plausible (Fig. 1).

The additional measurements left two potential explanations for the elevated concentrations; unidentified systematic measurement issues or contamination of sample material. Repeat measurements of the quartz interlaboratory comparison

material CRONUS-A (Jull et al., 2015; Goehring et al., 2019) and other samples allowed us to rule out systematic measurement issues and conclude that there must be an unidentified source of [14]C contamination. Measurements presented in Figs. 1 and 2 were made using quartz which was not only visually pure but had initially been isolated for [10]Be measurements. The samples had previously been sent for ICP-MS analysis to test their suitability for [10]Be analysis, confirming that they were comprised of sufficiently pure quartz and thus were ready for [14]C analysis as well. We are therefore confident that the elevated [14]C concentrations were not sourced from other minerals that persisted through quartz isolation. We suspected that the froth flotation procedure was a potential source of [14]C contamination because it involves the introduction of carbon to sample material through the use of three aforementioned compounds. We focused on the long-chain compound laurylamine because eucalyptol is volatile at room temperature and is thus unlikely to persist through sample etching. Acetic acid is predominantly sourced from methanol which is, in turn, largely derived from [14]C dead natural gas, though it can be produced using modern material and therefore may have the potential to contaminate samples with [14]C. However, regardless of the source, acetic acid is a simple compound that would be relatively easy to break down during etching when compared to laurylamine. There is a complicating factor, in that acetic acid and laurylamine can form complex molecules that behave as a singular species (Karlsson et al., 2001), which may increase the potential for acetic acid to remain on sample material after froth flotation and contribute to potential [14]C contamination. Again, though the predominantly [14]C dead source material minimises potential acetic acid influence. Nonetheless, we focused on laurylamine but acknowledge that it may not be the sole contributor to residual [14]C following froth flotation. The potential of laurylamine to contaminate in situ [14]C concentrations depends on the carbon source of the compound. With a modern source of carbon, laurylamine has the potential to introduce large quantities, relative to the in situ component, of [14]C to samples. The observed changes in [14]C concentration (seemingly dependent on where quartz was isolated and potentially the differing procedures used to isolate quartz) necessitated a systematic study into the potential source and scale of contamination and, if possible, how to efficiently and reliably remove it.

**2.0 Systematic Investigation**

We isolated quartz from a whole rock sample using five different methods in order to investigate the cause of contamination. The sample selected for this purpose is Caledonian trondhjemite bedrock (Ragnhildstveit et al., 1998) from Utsira, Norway. The sample contains significant feldspar, mica and quartz, making it ideal for use with froth flotation. The [14]C concentration of the sample is irrelevant for the present study; what is important is the ability to observe any potential contamination from the froth flotation procedure. Prior to froth flotation, the sample was crushed, milled, sieved (to isolate the 250 - 500 μm fraction), and magnetically separated. Following magnetic separation, quartz was isolated for aliquot 1 without froth flotation via four days on a shaker table in 5 % HF/HNO₃ followed by two days in an ultrasonic bath in 1 % HF/HNO₃. The ultrasonic bath is not heated, but through continued use reaches ca. 40 °C. Aliquot 1 thus forms a baseline against which the other aliquots are compared. Froth flotation was used with aliquots 2 to 5, which were then etched with different acid mixtures (HF and HF/HNO₃), and varied agitation methods (shaker table and an ultrasonic bath; Table 1). Aliquot 2 spent two

days on the shaker table in 5 % $HF/HNO_3$ and two days in the ultrasonic bath in 1 % $HF/HNO_3$, which is the minimum duration of etching that all samples receive in our laboratory. Aliquot 3 also spent two days on the shaker table and two days in the ultrasonic bath but was etched in only HF (5 % on the shaker table and 1 % HF in the ultrasonic bath). Aliquot 3 is essentially our standard procedure but without the inclusion of $HNO_3$. Aliquots 4 and 5 were not etched on the shaker table and both spent two days in the ultrasonic bath, after which they were visually pure, with the former etched in 1 % $HF/HNO_3$, and the latter etched in 1 % HF. Etching samples until quartz is visually pure is a common procedure used to isolate quartz for cosmogenic nuclide analysis. Aliquots 4 and 5 thus represent a feasible minimum duration of etching and were analysed to test if the short duration is sufficient to remove potential contamination. A new acid mixture was used with the samples following a set of rinses with ultrapure 18,2 MΩ water, such that each aliquot received a new acid mixture once every 24 hours.

We extracted carbon from the five quartz aliquots using the Tulane University Carbon Extraction and Graphitization System (TU-CEGS) following the method of Goehring et al. (2019). Quartz is step-heated in the presence of a lithium metaborate ($LiBO_2$) flux and a high-purity $O_2$ atmosphere, first at 500 °C for 30 minutes, then at 1100 °C for three hours. The former step is to remove any adsorbed atmospheric $CO_2$ and combust any carbon derived from sample handling. Released carbon species from the latter 1100 °C step are oxidised to form $CO_2$ via secondary hot-quartz-bed oxidation. This is followed by cryogenic collection and purification of the $CO_2$. Sample yields are measured manometrically (Table 2), and samples are diluted with $^{14}C$-free $CO_2$. A small aliquot of $CO_2$ is collected for $\delta^{13}C$ analysis, and the remaining $CO_2$ is graphitised using $H_2$ reduction over an Fe catalyst (e.g. Southon, 2007). Cathodes containing the graphite were sent to the Woods Hole National Ocean Sciences Accelerator Mass Spectrometry (NOSAMS) to measure $^{14}C/^{13}C$ isotope ratios (Table 2) relative to NIST SRM4990c Oxalic Acid II primary standard. The primary standard was produced in the same graphite reactors used for the unknowns, ensuring full internal normalisation. Stable carbon isotope ratios were measured at the UC-Davis Stable Isotope Facility (Table 2). Repeat measurements of the CRONUS-A interlaboratory comparison standard (Jull et al., 2015) and other samples using the TU-CEGS show that the reproducibility of in situ $^{14}C$ measurements is approximately 6 % (Goehring et al., 2019). We therefore present our $^{14}C$ concentrations with a conservative 6 % uncertainty as this exceeds the reported analytical uncertainty for all of our $^{14}C$ measurements. Typical total analytical uncertainties are 1.5 to 2.5 %. Blank corrections as a percentage of the total $^{14}C$ atoms in each sample range from 13.5 to 17.0 % (Table 2).

We also measured the carbon isotope ratio of laurylamine to both identify the presence of a modern carbon source for our laurylamine, and to permit a mass balance calculation to quantify the amount of laurylamine left behind after the frothing and etching process. We extracted carbon from 1.9 mg of laurylamine using the TU-CEGS. We used the process regularly used in our laboratory to extract carbon from oxalic acid. This was appropriate given the similarity of the decomposition temperatures of oxalic acid (~189 °C) and laurylamine (~178 °C). We combusted the sample at 150 °C for ten minutes in ~0.2 MPa (or ~2 atm) of ultra-high purity $O_2$, after which the temperature was increased to 500 °C to ensure complete combustion. The resulting $CO_2$ was then cryogenically collected and purified, followed by catalytic reduction via $H_2$ to graphite. As with the five quartz aliquots, the cathode was sent to NOSAMS to measure the $^{14}C/^{13}C$ isotope ratio relative to NIST SRM4990c Oxalic Acid II primary standard.

## 3.0 Results

Firstly, the fraction modern (Fm) value of laurylamine is $1.0338 \pm 0.0020$, indicative of a modern carbon source. Results for the five aliquots are shown in Table 2, with the unit yields and $^{14}C$ concentrations also presented in Fig. 3. The total carbon yields for aliquots 1 and 2 are lower than those of aliquots 3 to 5. Aliquots 1 and 2 were isolated without froth flotation and with the TUCNL standard procedure (including froth flotation), respectively. Because slightly differing masses of quartz were used for in situ $^{14}C$ analysis, a direct comparison can be made using the carbon unit yields (Fig. 3 and Table 2). The unit yield for aliquots 1 and 2 are the same within $1\sigma$ uncertainty. We observe elevated unit yields for aliquots 3 to 5 relative to those of aliquots 1 and 2.

As with the unit yields, the $^{14}C$ concentration of aliquots 1 and 2 are the same within uncertainties and are distinguishable from the unit yields of aliquots 3 to 5 when using the conservative 6 % uncertainty (Fig. 3). We observe elevated $^{14}C$ concentrations for aliquots 3 to 5 relative to those of aliquots 1 and 2, with a particularly high $^{14}C$ concentration for aliquot 5 (Fig. 3B). Figure 3 shows that the higher unit yields correspond with higher measured $^{14}C$ concentrations. With aliquot 5, a small increase in unit yield results in a disproportionately high $^{14}C$ concentration that dwarfs those of aliquots 1 to 4.

## 4.0 Discussion

The modern carbon source, identified with the measured Fm of our laurylamine, shows that laurylamine is not $^{14}C$ dead and thus has the potential to contaminate samples with respect to $^{14}C$. We did not measure the Fm of acetic acid or eucalyptol due to the rationale described above (Sect. 1.2) and thus we cannot rule out their potential to contaminate samples with $^{14}C$. However, the modern carbon source of laurylamine confirms that the froth flotation procedure, regardless of the contributing compound, introduces $^{14}C$ to sample material. The measured $^{14}C/^{12}C$ ratio for laurylamine is $1.19 \times 10^{-12}$. This means that, for example, 20 µg contains $\sim 9.3 \times 10^5$ atoms of $^{14}C$. The elevated carbon yields and unit yields of aliquots 3 to 5 relative to those of aliquots 1 and 2 may indicate that the former are contaminated with total carbon and, of particular importance, $^{14}C$. However, elevated carbon yields and unit yields are not sufficient evidence alone to indicate contamination because the maximum difference in carbon yields (2.2 µg, Table 2) is within the range of carbon yields of process blanks in our laboratory (Goehring et al., 2019). Therefore, the differing yields may simply be the result of varying blank magnitude and not due to contamination from froth flotation. However, the elevated $^{14}C$ concentrations of aliquots 3 to 5 relative to those of aliquots 1 and 2 do indicate that the former are contaminated with $^{14}C$. The difference in $^{14}C$ concentration between aliquots 1 and 2 and those of aliquots 3 to 5 is much greater than the $^{14}C$ content of process blanks in our laboratory (Goehring et al., 2019), therefore the difference cannot be explained by varying blank magnitudes alone and is indicative of $^{14}C$ contamination. The elevated unit yields may therefore also be due to carbon contamination. The unit yields and $^{14}C$ concentrations of aliquots 1 and 2 are indistinguishable from one another, which indicates that our standard procedure for quartz isolation (Aliquot 2) removes carbon introduced by laurylamine. Differing quartz isolation procedures used at other laboratories may therefore

explain why quartz isolated from the same samples at Tulane and elsewhere produced vastly different [14]C concentrations (Sect. 1.2).

We use the excess measured [14]C atoms in aliquots 3 to 5 (the total [14]C atoms for each aliquot in excess of the average of those of aliquots 1 and 2) with the measured [14]C/[12]C ratio for laurylamine to calculate the corresponding mass of residual carbon and laurylamine, per gram of quartz, that was collected with the in situ [14]C component. We assume aliquots 1 and 2 were not contaminated with [14]C and thus excess [14]C is sourced solely from laurylamine, though it could be sourced from eucalyptol or acetic acid if they were to persist through sample etching. To calculate the mass of contaminant ($M_{contam}$) carbon or laurylamine we follow

$$M_{contam} = {}^{14}C_{excess} \left( \frac{{}^{14}C}{{}^{12}C} \right)_{LA} \frac{M}{A}$$

where $^{14}C_{excess}$ is the measured number of excess [14]C atoms, $(^{14}C/^{12}C)_{LA}$ is the measured ratio for laurylamine (1.19 x 10$^{-12}$), $M$ is the molar mass of carbon or molecular mass of laurylamine, and $A$ Avogadro's Number. This calculation is an estimate as it does not take into account the ca. 1.1 % [13]C in laurylamine. The excess [14]C accounts for an estimated 0.06, 0.03 and 0.74 µg carbon g$^{-1}$ quartz, and 0.08, 0.04, 0.95 µg g$^{-1}$ of laurylamine for aliquots 3, 4 and 5, respectively (Table 2). We can use the same method for the samples that produced the initial anomalous measurements shown in Fig. 1. To do so we assume that the final measurement made for each sample is free from laurylamine contamination. For the samples presented in Fig. 1, the excess [14]C concentrations range from 1.38 x 10$^5$ to 3.23 x 10$^5$ at g$^{-1}$. The associated residual carbon ranges from 2.32 to 5.42 µg g$^{-1}$, and the residual laurylamine ranges from 2.98 to 6.96 µg g$^{-1}$, both per gram of quartz. We speculate that the latter residual carbon and laurylamine estimates, an order of magnitude greater than those presented in this study, may be an artefact of the differing froth flotation and etching procedures used at the laboratories at which the quartz was isolated. Contributing factors could include, but are not limited to, a greater amount of laurylamine used in the quartz separation process, the concentration at which the laurylamine comes into contact with sample material (dilute or undilute), the acids used in the etching procedure, and the duration of acid etching. As noted in Sect. 1.2, we are confident that the elevated [14]C concentrations were not sourced from other minerals that persisted through quartz isolation because the quartz separates were previously analysed by ICP-MS to confirm their suitability for [10]Be analysis. Though fluid inclusions may contribute to elevated carbon yields, they would presumably be devoid of [14]C and thus could not explain the anomalous [14]C concentrations. Production of [14]C on [14]N in fluid inclusions through thermal neutron capture is possible, however, the presumably low abundance of [14]N means that this production mechanism is unlikely to contribute significantly to [14]C concentrations when compared to the spallation component (Lal and Jull, 1998).

The elevated carbon yields and [14]C concentrations of aliquots 3 to 5 relative to those of aliquots 1 and 2 suggest two things. Firstly, it is apparent that HNO₃ is needed to remove laurylamine-derived carbon, both total carbon and [14]C, contamination from quartz. The importance of HNO₃ is demonstrated by the higher unit yields and [14]C concentrations of

aliquots that were etched with only HF compared to aliquots that had the same quartz isolation method and duration of etching but were etched with a combination of HF and $HNO_3$ (aliquot 3 vs 2 and aliquot 5 vs 4). We hypothesise that, as an oxidiser, $HNO_3$ is key in the decomposition of laurylamine. Before carbon is extracted from aliquots, quartz is leached in 50 % v/v $HNO_3$ for 30 minutes in an unheated ultrasonic bath (Lifton et al., 2001; Goehring et al., 2019). This is important to note because it is apparent that this additional leach with strong $HNO_3$ is not sufficient alone to remove contaminant [14]C and highlights the importance of HF as well as $HNO_3$ in the etching procedure and their role in the removal of contamination. We hypothesise that dissolution of quartz using HF helps to release contamination stored within microfractures of quartz grains (elaborated further below). Secondly, two days in an ultrasonic bath with 1 % acid mixture, regardless of whether HF or $HF/HNO_3$ is used for etching, appears to be insufficient to remove froth flotation-derived contaminants. Aliquots 4 and 5, which were not etched on the shaker table and spent a total of two days etching in an ultrasonic bath, both appeared visually pure and thus looked ready for in situ [14]C analysis without the context of potential froth flotation-derived contamination. It is possible that that contamination would have been removed if aliquots 4 and 5 were etched with 5 % rather than 1 % $HF/HNO_3$. However, the purpose of aliquots 4 and 5 was to test if the minimum feasible duration of etching and strength of acid used by laboratories to isolate quartz would be sufficient to remove potential contamination from froth flotation. Evidently, a standard procedure to etch samples until they are visually pure is not necessarily sufficient when froth flotation has been used. If a laboratory has only a shaker table or an ultrasonic bath, we would speculate that a minimum of four 24-hour periods in 5 % $HF/HNO_3$ would be sufficient to remove froth flotation-derived contamination.

The observation that the [14]C concentration increase from froth flotation is of the same order of magnitude as that of typical in situ [14]C measurements is of great concern and highlights the need for a sufficiently thorough minimum procedure to eliminate contamination from the quartz isolation process. Carbon introduced by froth flotation is evidently persisting through the 500 °C step heat, the first stage of extracting [14]C from quartz with the TU-CEGS and in other in situ [14]C laboratories (e.g., Hippe et al., 2013; Lifton et al., 2015; Goehring et al., 2019; Lamp et al., 2019). The 500 °C bake was previously shown to remove contaminant [14]C (Lifton et al, 2001), though this was presumably from sample handling and the atmosphere and pre-dates the implementation of froth floatation for quartz separation. We suspect that the observed contamination is sourced from laurylamine or other froth flotation-derived contaminants residing within microfractures of quartz grains, which may explain why the contamination is able to persist through the 500 °C bake and possibly accounts for differences in the degree of contamination between previously analysed samples and those as part of this study due to differences in the quartz grain characteristics. The potential quantity of laurylamine or other contaminants able to reside in the microfractures of a particular sample will presumably vary with the lithology and geologic history of the sample, as well as the methods of sample preparation. The natural abundance of microfractures in a sample prior to sample collection will vary and microfractures may also be introduced during sample collection, crushing and milling. Figures 4 and 5 show evidence of microfractures on the surface of quartz grains from all aliquots. In addition, Fig. 4 shows a quartz grain from an unetched aliquot that was sourced from the same whole rock sample as the five aliquots. Anecdotally, whilst using the SEM we observed microfractures that seemed to be opened up to a greater extent in aliquots 1 and 2, which received the longest duration of etching, compared to

aliquots 3 to 5. Note the high surface roughness of the unetched sample (Fig. 4a and b) and the relative smoothness of the grains in all aliquots (Figs. 4 and 5), a result of the partial dissolution by HF of quartz grains which will have presumably removed some microfractures entirely. We observe that further etching, both in our initial measurements (Sect. 1.2) and when comparing aliquots 2 and 3 with aliquots 4 and 5, lowers carbon yields and $^{14}C$ concentrations. The longer duration in acid may indicate that the HF is opening up microfractures and allowing contamination to be more thoroughly removed, highlighting the importance of HF in the removal of contamination, though this would be difficult to test, and an extensive systematic study would be required to make conclusions with any statistical significance. Whilst the presence of microfractures does not confirm our hypothesis, Figs. 3 and 4 do show that there are abundant microfractures and surface features for contaminants to potentially reside in following froth flotation.

## 5.0 Conclusion

We found that laurylamine is manufactured with a modern carbon source and thus introduces modern $^{14}C$ to sample material during froth flotation. We have shown through a systematic study that contaminant $^{14}C$ from froth flotation persists through sample etching and is collected with in situ $^{14}C$ if etching is not rigorous enough. Nitric acid, combined with hydrofluoric acid, is required to effectively remove contaminant $^{14}C$, which is shown by the elevated $^{14}C$ concentrations of quartz separates isolated without nitric acid relative to those extracted with nitric acid. We have outlined a reliable method for ensuring no contaminant $^{14}C$ from froth flotation remains with quartz following etching. In short, two 24-hour periods on a shaker table with 5 % HF/HNO$_3$, followed by two 24-hour periods in an ultrasonic bath in 1 % HF/HNO$_3$, is sufficient to produce in situ $^{14}C$ concentrations indistinguishable from a sample for which quartz was isolated without froth flotation. Ultimately, froth flotation should be used with caution and the sample etching procedure outlined above should be used at an absolute minimum.

## Data availability

All data are included with this manuscript.

## Author contributions

The study was conceived by BG and KN. Sample collection, preparation and analysis undertaken by KN and BG. Manuscript was written by KN and BG.

## Competing interests

The authors declare that they have no conflict of interest.

## Acknowledgements

KN and BG would like to acknowledge NOSAMS for their outstanding measurements. We would also like to acknowledge the fruitful discussions with Trevor Hillenbrand, John Stone, and Greg Balco that greatly helped to develop the study. Sample material was collected with the help of Jan Mangerud and John-Inge Svendsen. We would like to thank Jibao He for training and assistance in the use of the SEM in the CIF Microscopy Laboratory at Tulane University. We would also like to thank Anthony Jull and an anonymous reviewer for their reviews that helped to improve the manuscript.

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

## Figures

Figure 1: Elevation versus in situ $^{14}$C concentration of samples that initially yielded anomalously high $^{14}$C concentrations. Measurements at the same elevation are from the same sample. For each sample, the highest in situ $^{14}$C concentration is sourced from the first measurement (red). For the two samples measured three times at 510 and 875 m a.s.l., the intermediate measurement was made following additional etching and yields the intermediate in situ $^{14}$C concentration (yellow). For every sample, the final measurement (blue) was made from quartz isolated from whole rock at Tulane. Error bars reflect a long-term 6 % uncertainty. Some error bars are smaller than their respective data points. Thick grey line and grey shading are the saturation concentration and associated error envelope.

Figure 2: Initial and final unit yields associated with the same measurements presented in Fig. 1. The initial unit yield measurements for each sample were made using quartz isolated at external laboratories, whilst the final unit yield measurements were made using quartz isolated at Tulane using our standard procedure. Error bars are smaller than the data points.

Figure 3: A: Unit yield for the five quartz separates. Aliquot numbers refer to those in Table 1. B: C-14 concentration for the five quartz aliquots; error bars reflect a long-term 6 % uncertainty. B has a split y-axis to present both the differences in $^{14}$C concentrations between aliquots 1 to 4 and the difference between aliquot 5 and aliquots 1 to 4. See Table 1 for the different quartz isolation procedures used. For reference, all aliquots other than Aliquot 1 were subject to froth flotation. Aliquot 2 was processed using the TUCNL standard procedure.

Figure 4: SEM images of quartz grains of an unetched sample and aliquots 1 and 2. Red boxes on the left show the location of the adjacent image to the right. The unetched sample is sourced from the same whole rock sample as the five aliquots and was crushed, milled, sieved, rinsed and magnetically separated. Note the conchoidal fracture in B.

Figure 5: SEM images of quartz grains of aliquots 3 to 5.

## Tables

Table 1: Aliquot information and quartz isolation procedures. Whilst on the shaker table, samples were etched in 5 % HF/HNO$_3$ or 5 % HF. Whilst in the ultrasonic bath, samples were etched in 1 % HF/HNO$_3$ or 1 % HF.

Table 2: Table 2: In situ $^{14}$C analytical data. Aliquot number is described in the text. See Table 1 for the different quartz isolation procedures used for each aliquot. TUCNL is a unique sample identifier for each sample analysed at the Tulane University Cosmogenic Nuclide Laboratory. C yield is the carbon yield prior to dilution. Unit yield is the carbon yield divided by the quartz mass. Total $^{14}$C blank corrected is the total number of $^{14}$C atoms corrected using the effective blank. Effective blank is representative of the blank during the running of the samples presented. See Sect. 2 for rationale behind the use of the 6 % uncertainty for the $^{14}$C concentrations. We also include 1σ uncertainty for the $^{14}$C concentrations for completeness. The mass of residual carbon and laurylamine for aliquots 3 to 5 are calculated using the $^{14}$C/$^{12}$C ratio of laurylamine as measured (see Sect. 4)

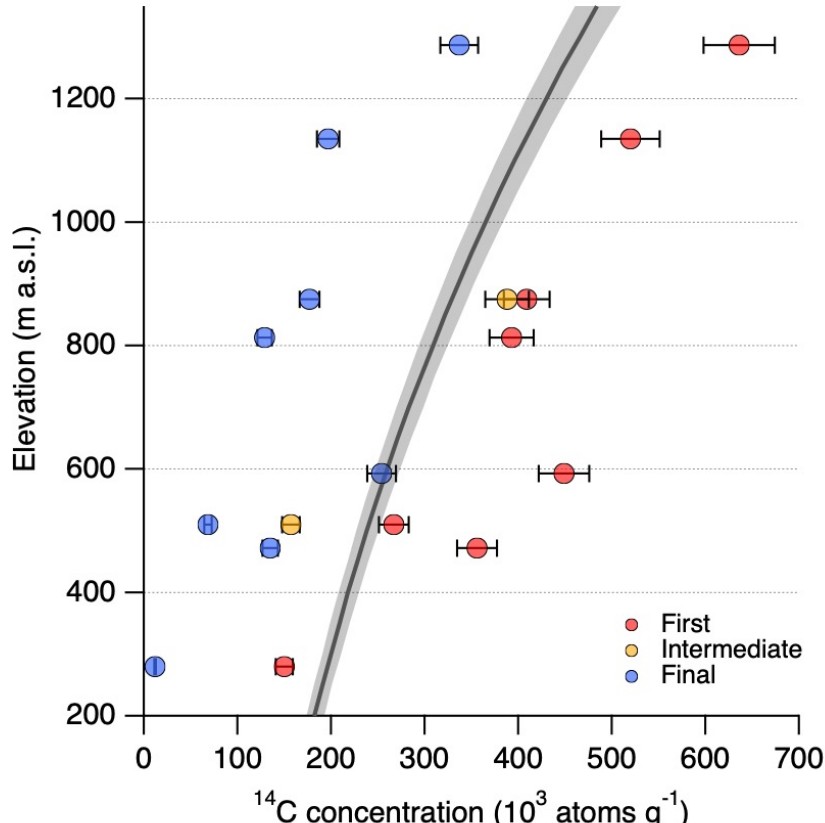

Fig. 1

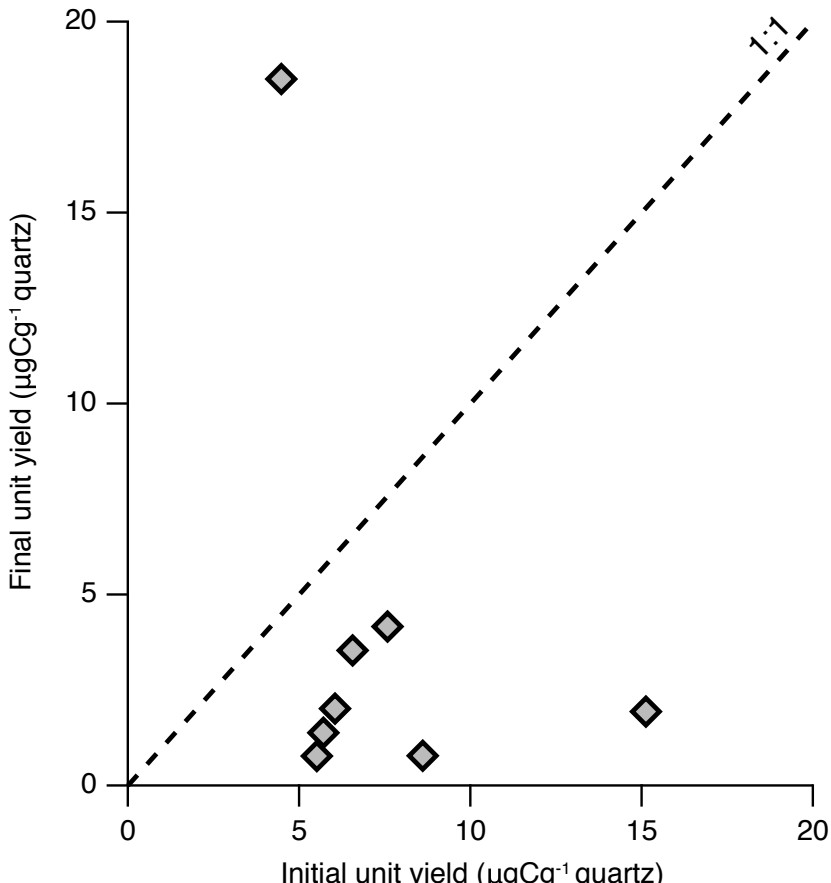

Fig. 2



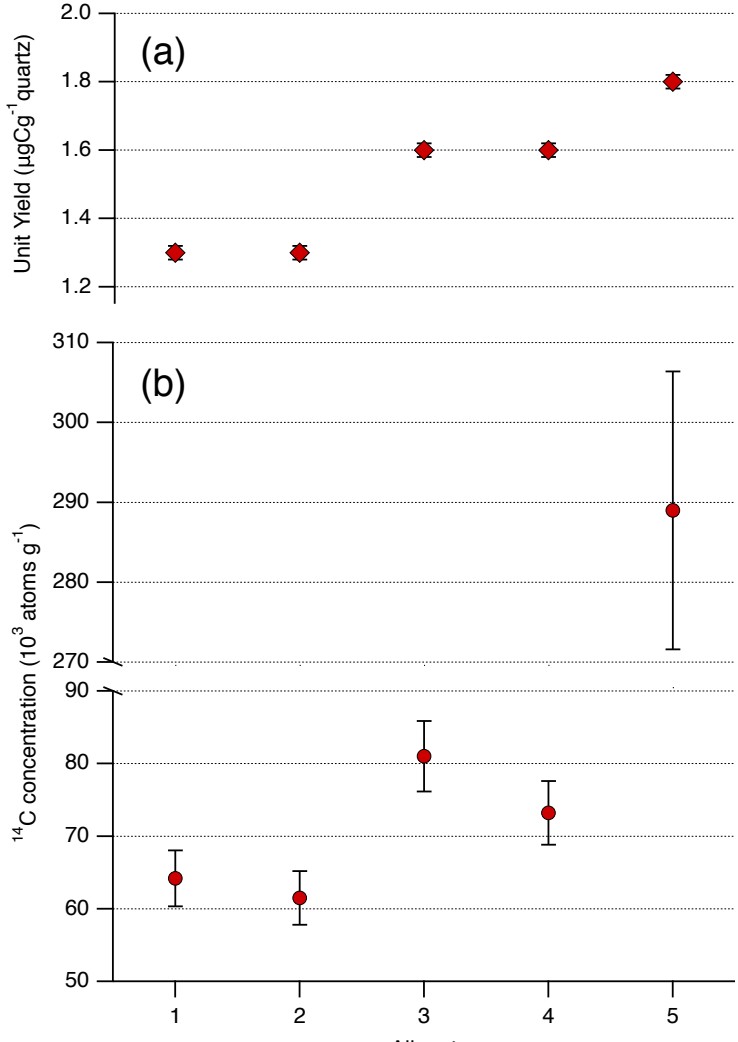

Fig. 3

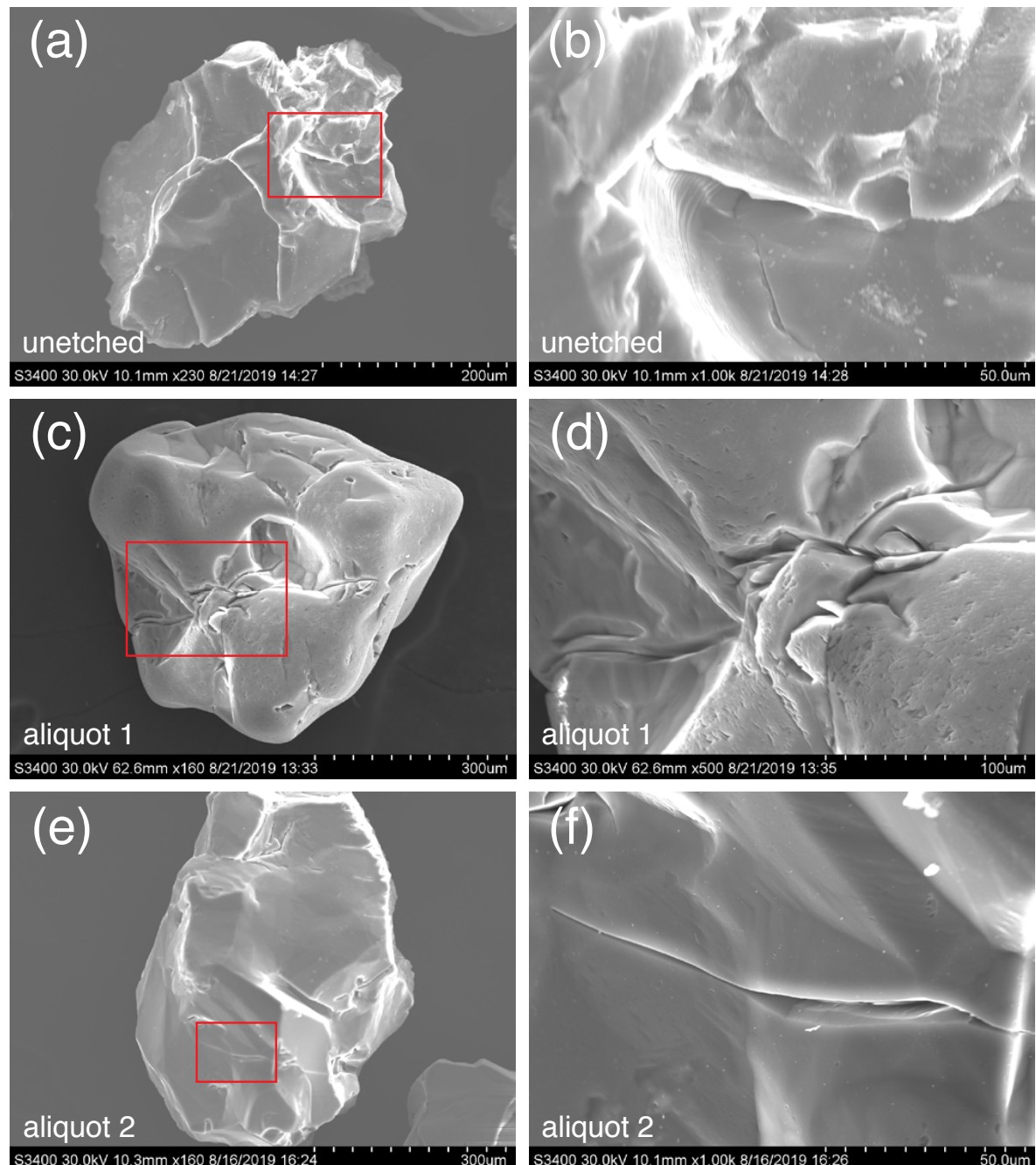

Fig. 4

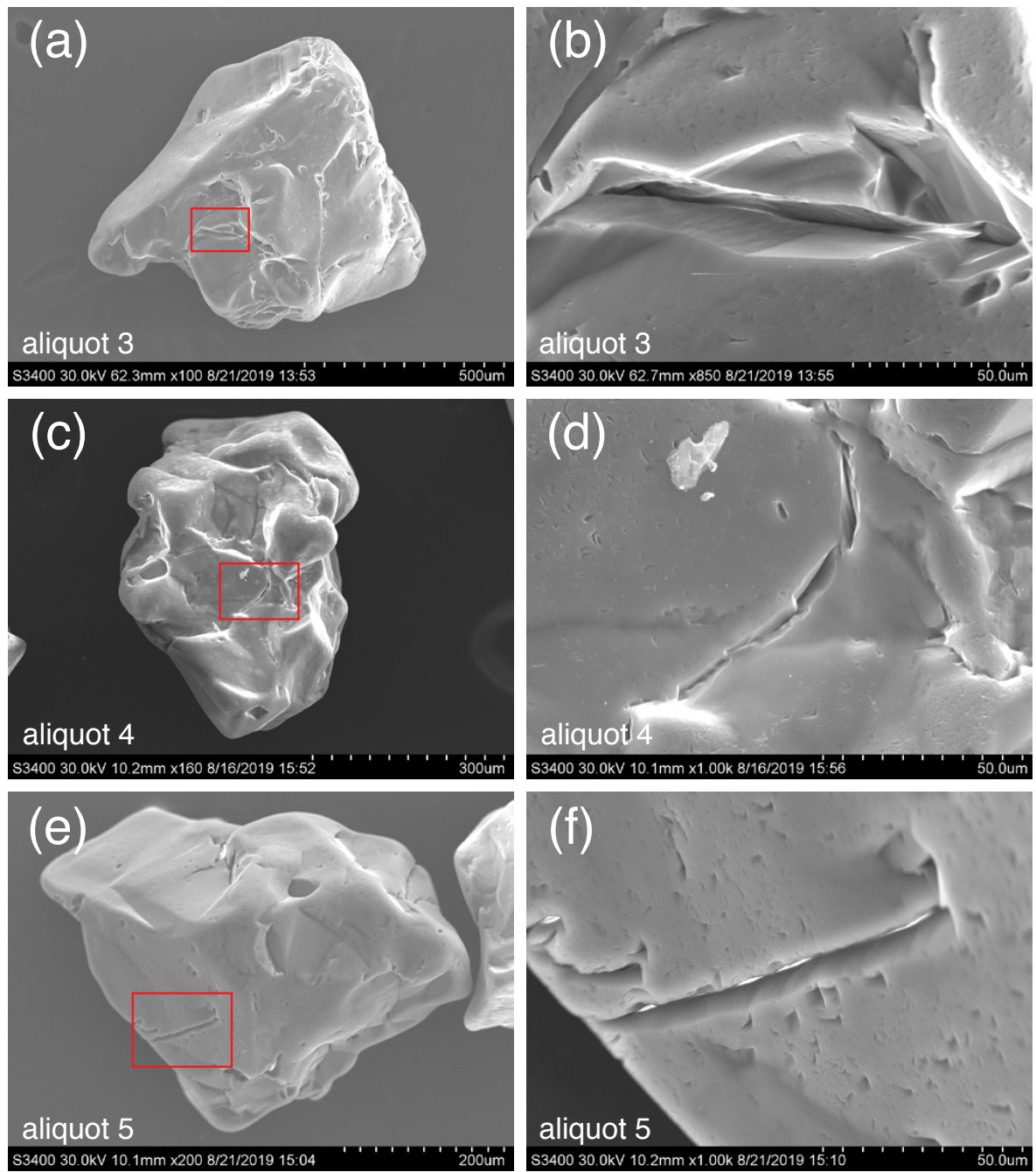

Fig. 5

| Sample ID | Aliquot Number | Days on shaker table | Days in ultrasonic bath | Total | Notes |
|---|---|---|---|---|---|
| 16-UT-004-QUA-NOFROTH | 1 | 4 | 2 | 6 | No froth flotation, etched with HF/HNO$_3$ |
| 16-UT-004-QUA-NORM | 2 | 2 | 2 | 4 | Froth flotation, etched with HF/HNO$_3$ in shaker table and ultrasonic bath |
| 16-UT-004-QUA-HFONLY | 3 | 2 | 2 | 4 | Froth flotation, etched with HF in shaker table and ultrasonic bath |
| 16-UT-004-QUA-NOSTABLE1 | 4 | 0 | 2 | 2 | Froth flotation, etched with HF/HNO$_3$ in ultrasonic bath |
| 16-UT-004-QUA-NOSTABLE12 | 5 | 0 | 2 | 2 | Froth flotation, etched with HF in ultrasonic bath |

Table 1.




| Aliquot Number | TUCNL | AMS Lab | AMS ID | Quartz weight (g) | C yield ($\mu$g) | ±1$\sigma$ ($\mu$g) | Unit Yield ($\mu$g g$^{-1}$) | Diluted Gas Mass ($\mu$g) | ±1$\sigma$ ($\mu$g) |
|---|---|---|---|---|---|---|---|---|---|
| 1 | 309 | NOSAMS | OS-141782 | 5.196 | 6.9 | 0.1 | 1.3 | 85.5 | 1.1 |
| 2 | 307 | NOSAMS | OS-141784 | 5.122 | 6.5 | 0.1 | 1.3 | 100.3 | 1.3 |
| 3 | 308 | NOSAMS | OS-141785 | 5.104 | 8.3 | 0.1 | 1.6 | 88.0 | 1.1 |
| 4 | 310 | NOSAMS | OS-141786 | 5.134 | 8.4 | 0.1 | 1.6 | 83.7 | 1.1 |
| 5 | 311 | NOSAMS | OS-141788 | 5.080 | 9.1 | 0.1 | 1.8 | 76.7 | 1.0 |



| $^{14}$C/$^{13}$C | ±1$\sigma$ | $\delta^{13}$C (‰) | ±1$\sigma$ (‰) | $^{14}$C/C total | ±1$\sigma$ | Total $^{14}$C atoms blank corrected | ±1$\sigma$ (at) | $^{14}$C conc. (at.g$^{-1}$) |
|---|---|---|---|---|---|---|---|---|
| 8.47E-12 | 8.56E-14 | -4.98 | 0.5 | 9.29E-14 | 9.39E-16 | 3.34E+05 | 9.439E+03 | 6.42E+04 |
| 6.88E-12 | 8.23E-14 | -4.54 | 0.5 | 7.55E-14 | 9.04E-16 | 3.15E+05 | 9.548E+03 | 6.15E+04 |
| 9.88E-12 | 9.58E-14 | -5.20 | 0.5 | 1.08E-13 | 1.05E-15 | 4.13E+05 | 1.029E+04 | 8.10E+04 |
| 9.57E-12 | 9.27E-14 | -5.13 | 0.5 | 1.05E-13 | 1.02E-15 | 3.76E+05 | 9.845E+03 | 7.32E+04 |
| 3.64E-11 | 1.90E-13 | -5.78 | 0.5 | 3.99E-13 | 2.08E-15 | 1.47E+06 | 2.230E+04 | 2.89E+05 |

| ±1$\sigma$ (at.g$^{-1}$) | ±6% (at.g$^{-1}$) | Effective blank (at) | ±1$\sigma$ (at) | Effective Blank as % of total $^{14}$C At Sample | Residual C ($\mu$g g$^{-1}$) | Residual C$_{12}$H$_{27}$N ($\mu$g g$^{-1}$) |
|---|---|---|---|---|---|---|
| 1.64E+03 | 3.85E+03 | 6.47E+04 | 6.85E+03 | 16.25 | - | - |
| 1.64E+03 | 3.69E+03 | 6.47E+04 | 6.85E+03 | 17.04 | - | - |
| 1.80E+03 | 4.86E+03 | 6.47E+04 | 6.85E+03 | 13.53 | 0.06 | 0.08 |
| 1.73E+03 | 4.39E+03 | 6.47E+04 | 6.85E+03 | 16.25 | 0.03 | 0.04 |
| 4.10E+03 | 1.74E+04 | 6.47E+04 | 6.85E+03 | 14.69 | 0.74 | 0.95 |


Table 2. (split into three sections)