# Peer review of "Isolation of quartz for cosmogenic in situ 14C analysis"

_Geochronology, 2019_

## Referee Comment (RC1) · Anonymous Referee #1 · 29 Jul 2019

Nichols and Goehring investigate the froth flotation technique used to separate feldspars from quartz as a potential source of contamination in in-situ C-14 analyses. The authors argue that residual laurylamine from the froth flotation process is a source of modern C-14 and thus will yield anomalously high measured C-14 concentrations, if residual laurylamine is not removed properly during the quartz purification process. The manuscript then provides guidelines on how to clean quartz for in situ C-14 analysis.

While I do think that the work presented in this manuscript is useful and of interest to the handful of in-situ C-14 labs, I find the current manuscript disappointing and deficient in many aspects.

(1) Eucalyptus oil

[Figure]

In addition to laurylamine and acetic acid, the froth flotation process also relies on eucalyptus oil (substituted in some labs with pine oil). Eucalyptol and terpineol, the main constituents of eucalyptus and pine oils respectively, are both organic compounds. Further, it is possible that during the frothing process, samples will come into contact with equal or more eucalyptus or pine oils than with laurylamine (for example in our lab we add 2 g dodecylamine to 20 L of H2O, but add generous amounts of eucalyptus oil every time the sample gets sprayed with the frothing mixture). I was wondering whether the authors of the study have investigated these compounds?

Furthermore, in line 165 the authors present the "complete combustion of laurylamine' at 500 oC arguing that its decomposing below 200 oC. If this is a valid statement I was wondering if the authors try to test the removal of extraneous carbon by adjusting the pre-heating step duration and temperature?

(2) Caledonian trondhjemite bedrock sample

I do understand very well why one would use a 'raw' rock sample for this experiment as opposed to for example using the CRONUS-A lab intercomparison material. The latter has already been purified and one needs a rock sample to go through the quartz purification process. Despite the authors arguing that the actual C-14 concentration of the trondhjemite bedrock sample is not important, I still believe that CRONUS-A, or a material with a known and confirmed C-14 concentration would have been a better choice. As the authors no doubt know, the extraction of in situ C-14 is still far from being routine. The handful of laboratories that exist use fairly different approaches to extract carbon and certain extraction system designs – such as the one TU-CEGS is based on – produce blanks that are one order of magnitude larger than say for example the ETH or Cologne/ANSTO systems. The authors might have a straight forward answer for this but looking at Table 3 in Goehring et al 2019 NIMB, blanks have C yields of between 13 and 1.9 ug. The maximum difference in C yield for this study (Table 2 of manuscript) is 2.2 ug between the 5 samples analysed – quartz masses are quite similar and so probably this does not have a large effect. Could the observed difference thus be

due to blank magnitude and variability rather than leftover laurylamine? For these reasons, it would have been nice to have some indication on what the expected C-14 concentration in the samples being used for the experiment, is. Would it be possible to estimate based on Be-10 or other information what the expected C-14 in this rock material would be? This would lend more credibility to the results presented here.

(3) Quartz isolation procedures

I would suggest a more careful formulation of the purpose of the froth floatation (referring to line 30) which never intended to replace density separation, and serves as a crude separation of feldspar minerals from quartz. Similarly, I was wondering whether there was any significance to the 'metal bowl' (line 80) used? Would a plastic bowl work?

Unfortunately, the manuscript does not provide information on how often the acid mixtures were changed during each of the steps. For example, in Table 1 during the 4 days on the shaker table (samples 1), was the acid mixture changed or the same HF/HNO3 was used for 4 days? This information would be useful if the authors wanted others to follow some of the recommendations provided. It should also be noted that some C-14 labs perform a concentrated HNO3 wash of the purified quartz at temperatures of 120 – 140 oC and this might well remove any residual laurylamine. At ETH, the HNO3 wash is followed by drying of the samples using an UV lamp. The authors note on line 220 that a HNO3 etch is performed but do not provide information on whether the samples are heated during the etch or not. Would this make a difference and did the authors look into that?

On a related note, the authors recommend the use of both shaker table and ultrasonic bath. Most labs will have one or the other and, again, it would have been useful – if the intention is to get people to adopt the recommendations presented here – to perform experiments for each of these (shaker table and ultrasonic bath) separately instead of changing the acid concentration between the two equipment. I am certain that if the

ultrasonic bath samples would have been etched with 5% acid mixture the result would be identical. It would also have been informative to present some ICP data on sample purity following the various steps.

Further, I would also assume that the amount of laurylamine (or eucalyptol) that could get trapped in cracks in the quartz grains will also depend on the type of sample and history of cleaning prior to froth flotation – i.e., some quartz grains will be more damaged than others.

(4) Figures

The figures presented in this manuscript need a bit more work. Figure 1 is especially difficult to read and the use of colour or different symbols would help the reader. Also using arrows to guide the eye as to the direction in which points should be shifting, would help. Also, should there be a table accompanying Figure 1? Or is this data published elsewhere? Figure 2: it is confusing, especially with the split into B and C.

(5) Table 2

Did the authors apply the same 6% uncertainty everywhere? The error on the number of atoms blank corrected is identical to the error on the effective blank. This simply cannot be.

(6) Technical comments

Line 150: 'to remove any adsorbed atmospheric CO2 and combust any carbon derived from handling and dust.' I would think that the removal of "dust" is entirely dependent on what is the dust made of and would be removed at 500 oC only if it is made of organic components. In most cases, however, dust is composed of inorganic particles.

Line 160: 'Typical total analytical uncertainties are 1.5 to 2.5 % including the blank correction.' I recommend that authors remove this statement as it is incorrect as this depends on the activity of the sample and also the relative blank contribution. For example, in Table 2 in the current manuscript the blank correction is ∼16%. Wouldn't

this have an effect on uncertainties?

Line 175: 'with the unit yields, the 14C concentration of aliquots 1 and 2 are the same within uncertainties and are distinguishable from the unit yields of aliquots 3 to 5 when using the conservative 6 % uncertainty (Fig. 2). We observe elevated 14C concentrations for aliquots 3 to 5 relative to those of aliquots 1 and 2, with a particularly high 14C concentration for aliquot 5 (Fig. 2B). Figure 2 shows that the higher unit yields correspond with higher measured 14C concentrations.' If this statement is correct, shouldn't the unit yield for aliquot 5 be 2.5 ugC?

Line 190 'Differing quartz isolation procedures used at other laboratories may therefore explain why quartz isolated from the same samples at Tulane and elsewhere produced vastly different 14C concentrations and unit yields (Sect. 1.2). ' Given the above points explained in detail I think this statement is only partially valid and would recommend a more careful explanation of the concentration differences.

Line 205-210; 'final measurement made for each sample is free from laurylamine contamination. For the samples presented in Fig. 1, the excess 14C concentrations range from 1.38 x 105 to 3.23 x 105 at g-1. The associated residual carbon ranges from 2.32 to 5.42 $\mu$g g-1, and the residual laurylamine ranges from 2.98 to 6.96 $\mu$g g-1, both per gram of quartz. We speculate that the latter residual carbon and laurylamine estimates, an order of magnitude greater than those presented in this study.' 7 ug of laurylamine /gram of quartz sounds like a large number. I was wondering whether the authors have considered other potential sources of the excess C. Perhaps it could be related to fluid inclusions or other minerals present in the sample that are only removed following additional HF leaching?

---

## Referee Comment (RC2) · Anthony Jull (Referee) · 29 Jul 2019

This paper presents some interesting observations about sources of 14C contamination during sample processing for in situ cosmogenic 14C measurements. A contamination from the "frothing process" using laurylamine is reported. I think this is an interesting study and deserves publication, however I have a few comments.

1. In lines 150-152, it is stated that the samples are diluted with (presumably dead) $CO_2$. In table 2, this appears to be corrected for the dilution, but the values given for 14C/13C appears to be 8.47 x 10-12 to 3.64 x 10-11 which must be incorrect. Modern carbon is about 10ˆ-10 14C/13C. The value stated in the paper for the laurylamine is 1.03 times modern (i.e. about 1.2 x 10-12 14C/12C. for 14C/13C this should be around

10ˆ-10), so even if the sample was 100% the contaminant this would still be wrong. I assume this is some arithmetic error but it needs to be corrected. 2. In table 2, an explanation of the various columns would be helpful. 3. In table 2, a value of d13C ca. -5 per mil is given. I assume this is of the diluted (not undiluted) gas? 4. The authors also note that the procedure involves adding the laurylamine to acetic acid. Yet, the acetic acid can be either from biogenic or nonbiogenic sources. Was this tested for 14C? 5. The authors might wish to review the chemistry of this process and the different phases that can form, for example there is a paper by S. Karlsson et al. (2001) Phase Behavior and Characterization of the System Acetic Acid-Dodecylamine-Water, Langmuir 17, 3573.

Sincerely, Timothy Jull (reviewer)

---

## Author Comment (AC1) · 6 Sep 2019

Please see the attached file for our responses to the comments of both reviewers.

Please also note the supplement to this comment:
https://www.geochronology-discuss.net/gchron-2019-7/gchron-2019-7-AC1-supplement.pdf

---

## Author Comment (AC2) · 6 Sep 2019

We would like to thank both reviewers for their thoughtful and thorough reviews that have helped significantly to develop the manuscript.

Text in blue, bold and italics are comments made by the reviewer, with our responses and changes to the text in regular text following.
* * *
**Reviewer 1**

*(1) Eucalyptus oil*
*In addition to laurylamine and acetic acid, the froth flotation process also relies on eucalyptus oil (substituted in some labs with pine oil). Eucalyptol and terpineol, the main constituents of eucalyptus and pine oils respectively, are both organic compounds.*

The reviewer raises a great point that there are two organic compounds used as part of the froth flotation procedure. We did not investigate acetic acid in our study. The rationale behind this is discussed below.

*Further, it is possible that during the frothing process, samples will come into contact with equal or more eucalyptus or pine oils than with laurylamine (for example in our lab we add 2 g dodecylamine to 20 L of H2O, but add generous amounts of eucalyptus oil every time the sample gets sprayed with the frothing mixture).*

The volume of eucalyptus oil in the above point seems, to our knowledge, to be unnecessarily high. Our method follows that of Purdue University's PRIME Lab, whom recommend using a few drops of eucalyptus oil every time the sample is mixed with the frothing solution, which has proven to be sufficient in our laboratory. We have seen documented comparable amounts of eucalyptus oil/pine oil at numerous other laboratories. As noted above, the reviewer raises an important point regarding the use of eucalyptus oil, which we address below.

*I was wondering whether the authors of the study have investigated these compounds?*

The rationale for focusing on laurylamine has now been added to the text, page 5, lines 154 to 165:

"We suspected that the froth flotation procedure was a potential source of $^{14}$C contamination because it involves the introduction of carbon to sample material through the use of three aforementioned compounds. We focused on the long-chain compound laurylamine because eucalyptol is volatile at room temperature and is thus unlikely to persist through sample etching. Acetic acid is predominantly sourced from methanol which is, in turn, largely derived from $^{14}$C dead natural gas, though it can be produced using modern material and therefore may have the potential to contaminate samples with $^{14}$C. However, regardless of the source, acetic acid is a simple compound that would be relatively easy to break down during etching when compared to laurylamine. There is a complicating factor, in that acetic acid and laurylamine can form complex molecules that behave as a singular species (Karlsson et al., 2001), which may increase the potential for acetic acid to remain on sample material after froth flotation and contribute to potential $^{14}$C contamination. Again, though the predominantly $^{14}$C dead source material minimises potential acetic acid influences. Nonetheless, we focused on laurylamine but acknowledge that it may not be the sole contributor to residual $^{14}$C following froth flotation."

We have altered the text to take a more general approach whenever referring to laurylamine contamination. We now state that laurylamine has the potential to contaminate samples with modern carbon, but we do not know if it is necessarily that specific compound (or another factor) contaminating samples. We state wherever needed that froth flotation in general is contaminating samples, not necessarily only laurylamine. Examples of this are:

Page 1, lines 14 to 18:

"Furthermore, we show that insufficient sample etching results in contaminant $^{14}$C persisting through step heating of quartz that is subsequently collected with the in situ component released at 1100 °C. We demonstrate that froth flotation contaminates in situ $^{14}$C measurements. We provide guidelines for the preparation of quartz based on

methods developed in our laboratory and demonstrate that all froth flotation-derived carbon and $^{14}$C is removed when applied."

Page 1, lines 22 to 25:

"We hypothesise that the elevated in situ $^{14}$C concentrations are sourced from part of the widely used mineral separation procedure known as froth flotation, a process that relies on three organic compounds; laurylamine (also known as dodecylamine, $C_{12}H_{27}N$), eucalyptol ($C_{10}H_{18}O$), and acetic acid ($C_2H_4O_2$)."

Page 2, lines 41 to 43:

"There is no standard procedure for froth flotation or post-froth flotation sample etching. As a result, different laboratories use various quantities of laurylamine, eucalyptol, and acetic acid…"

Page 2, lines 52 to 55:

"We conclude that froth flotation should be applied with care if in situ $^{14}$C is to be measured, and that the post-froth flotation etching methodology described below should be applied at a minimum to ensure that samples are free of contaminant $^{14}$C from froth flotation."

Page 7, lines 235 to 238:

"We did not measure the Fm of acetic acid or eucalyptol due to the rationale described above (Sect. 1.2) and thus we cannot rule out their potential to contaminate samples with $^{14}$C. However, the modern carbon source of laurylamine confirms that the froth flotation procedure, regardless of the contributing compound, introduces $^{14}$C to sample material."

Page 8, lines 262 to 264:

"We assume aliquots 1 and 2 were not contaminated with $^{14}$C and thus excess $^{14}$C is sourced solely from laurylamine, though it could be sourced from eucalyptol or acetic acid if they were to persist through sample etching."

Page 9, lines 310 to 314:

"The observation that the $^{14}$C concentration increase from froth flotation is of the same order of magnitude as that of typical in situ $^{14}$C measurements is of great concern and highlights the need for a sufficiently thorough minimum procedure to eliminate contamination from the quartz isolation process. Carbon introduced by froth flotation is evidently persisting through the 500 °C step heat, the first stage of extracting $^{14}$C from quartz with the TU-CEGS and in other in situ $^{14}$C laboratories (e.g., Hippe et al., 2013; Lifton et al., 2015; Goehring et al., 2019; Lamp et al., 2019)."

We believe that the results show that froth flotation contaminates samples with modern carbon, but we now note that the contamination is mostly likely sourced from one of the aforementioned compounds (acetic acid, eucalyptus/pine oil, and laurylamine), and give our reasons for thinking it is laurylamine that is the source (or main source). The take home message of the paper is that froth flotation contaminates samples with modern carbon. The post-froth flotation etching methods we use in our laboratory can be used to remove said contamination, which forms the second intended take home message of the paper.

*Furthermore, in line 165 the authors present the "complete combustion of laurylamine' at 500 oC arguing that its decomposing below 200 oC. If this is a valid statement I was wondering if the authors try to test the removal of extraneous carbon by adjusting the pre-heating step duration and temperature?*

We have indeed tried this, but it did not help with laurylamine, perhaps because it (or acetic acid and/or eucalyptol) is residing in microfractures. Furthermore, just because laurylamine decomposes sufficiently for $^{14}$C extraction to measure its $^{14}$C activity, does not mean it is removed completely during step heating.

*(2) Caledonian trondhjemite bedrock sample*
*I do understand very well why one would use a 'raw' rock sample for this experiment as opposed to for example using the CRONUS-A lab intercomparison material. The latter has already been purified and one needs a rock sample to go through the quartz purification process. Despite the authors arguing that the actual C-14 concentration of the trondhjemite bedrock sample is not important, I still believe that CRONUS-A, or a material with a known and confirmed C-14 concentration would have been a better choice.*

We did not want to use a sample of pure quartz with a known in situ $^{14}$C concentration because we do not know if froth flotation undertaken using pure quartz would be a fair representation of using froth flotation with a whole rock sample. We wanted to undertake the froth flotation procedure as one would for an in situ $^{14}$C study in its entirety. Essentially 'doping' a sample of pure quartz with laurylamine, followed by etching, may not have the same impact as using froth flotation with a whole rock sample and separating the feldspars and micas from the quartz. One hypothesis is that modern carbon is residing in microfractures following froth flotation and is being released either through further etching or during extraction of in situ $^{14}$C. Through etching samples that are already pure quartz (e.g. CRONUS-A), grains have presumably been rounded and microfractures may have been opened up. It therefore would not be a fair comparison to undertake froth flotation using pure quartz to test the impact of froth flotation with respect to contamination with modern carbon as opposed to using a whole rock sample that has not been etched previously.

We also had a number of reservations in the use of CRONUS-A in particular. As the reviewer will be aware, it is well established that CRONUS-A is saturated with respect to in situ $^{14}$C, meaning that is has a relatively high in situ $^{14}$C concentration (ca. $6 - 7 \times 10^5$ at g$^{-1}$). With a high concentration sample, it is more difficult to detect potentially elevated $^{14}$C concentrations resulting from contamination. Using a lower concentration sample means that the contamination is easier to detect.

Furthermore, there is a large range of measured values in the literature for the $^{14}$C concentration of CRONUS-A. For example, the range reported by Jull et al. (2015) based on 23 measurements from five chemistry laboratories and four AMS laboratories is $6.51 \pm 0.33 \times 10^5$ to $7.25 \pm 0.36 \times 10^5$ atoms g$^{-1}$. The range of reported values in increased when taking into account measurements made in our laboratory ($6.12 \pm 0.32 \times 10^5$ atoms g$^{-1}$ n=13; Goehring et al., 2019) as well as those reported by Lupker et al. (2019) ($7.27 \pm 0.03 \times 10^5$ atoms g$^{-1}$ (n=7)). The total range in the literature is thus ca. $1.15 \times 10^5$ atoms g$^{-1}$. The relatively large amount of scatter in published CRONUS-A measurements is greater than the excess $^{14}$C concentration of aliquots 3 and 4 ($1.95 \times 10^4$ and $9.01 \times 10^3$ atoms g$^{-1}$, respectively) and of the same order of magnitude of the excess $^{14}$C concentration of aliquot 5 ($2.25 \times 10^5$ atoms g$^{-1}$). The degree of scatter in CRONUS-A measurements either vastly exceeds or is at least of the same order of magnitude of the measured excess $^{14}$C in the aliquots presented in our study. Were the same magnitudes of excess $^{14}$C observed in measurements of CRONUS-A, it would have been difficult to say with confidence that the changes in $^{14}$C concentration in our study were due to $^{14}$C contamination or were simply due to scatter in the measurements of CRONUS-A. The cause or causes of excess scatter in the reported CRONUS-A measurements will take further work to identify. We believe the excess scatter was enough reason to not use CRONUS-A for this present study.

Additionally, we have observed a higher degree of scatter when measuring the in situ $^{14}$C content of samples derived from sandstones (such as CRONUS-A, but not limited to it) compared with measurements made from granitic sample material. This is why we went with the trondhjemite bedrock sample for this study. Another reason was a surplus of material of this particular sample, so using the trondhjemite bedrock sample had the added benefit of being cost-effective.

*As the authors no doubt know, the extraction of in situ C-14 is still far from being routine. The handful of laboratories that exist use fairly different approaches to extract carbon and certain extraction system designs – such as the one TU-CEGS is based on – produce blanks that are one order of magnitude larger than say for example the ETH or Cologne/ANSTO systems.*

According to Lupker et al. (2019), whom detail the ETH system, and Fülöp et al. (2018), whom detail the ANSTO system, the above statement is not correct. The reported procedural blank from Lupker et al. (2019) from their most recent set of measurements is $1.94 \pm 0.56 \times 10^4$ at. In Fülöp et al. (2018), procedural blanks using synthetic graphite are reported to be $\sim 1 \times 10^4$ at. The effective blank shown in Table 2 is $6.47 \times 10^4$ atoms, and the average of the two procedural blanks run directly before and after the samples presented in this study is $8.09 \times 10^4$ at. Whilst higher than the recently published procedural blanks of the ETH and ANSTO systems, this is not an order of magnitude higher. We would argue, and Lupker et al. and Fulop et al would likely agree, that extraction of in situ $^{14}$C has become much more routine in the last few years, but we are unsure how the above comment from the reviewer adds to the review.

*The authors might have a straight forward answer for this but looking at Table 3 in Goehring et al 2019 NIMB, blanks have C yields of between 13 and 1.9 ug. The maximum difference in C yield for this study (Table 2 of manuscript) is 2.2 ug between the 5 samples analysed – quartz masses are quite similar and so probably this does not have a large effect. Could the observed difference thus be due to blank magnitude and variability rather than leftover laurylamine?*

We agree that blank variability will play a role, to an extent, in the differing carbon yields and have added it to the text. Page 7, lines 238 to 248 now reads:

"This means that, for example, 20 µg contains $\sim 9.3 \times 10^5$ atoms of $^{14}$C. The elevated carbon yields and unit yields of aliquots 3 to 5 relative to those of aliquots 1 and 2 may indicate that the former are contaminated with total carbon and, of particular importance, $^{14}$C. However, elevated carbon yields and unit yields are not sufficient evidence alone to indicate contamination because the maximum difference in carbon yields (2.2 µg, Table 2) is within the range of carbon yields of process blanks in our laboratory (Goehring et al., 2019). Therefore, the differing yields may simply be the result of varying blank magnitude and not due to contamination from froth flotation. However, the elevated $^{14}$C concentrations of aliquots 3 to 5 relative to those of aliquots 1 and 2 do indicate that the former are contaminated with $^{14}$C. The difference in $^{14}$C concentration between aliquots 1 and 2 and those of aliquots 3 to 5 is much greater than the $^{14}$C content of process blanks in our laboratory (Goehring et al., 2019), therefore the difference cannot be explained by varying blank magnitudes alone and is indicative of $^{14}$C contamination. The elevated unit yields may therefore also be due to carbon contamination."

Though the blanks in Goehring et al. (2019) have carbon yields of between 13.5 and 1.9, the range of the carbon yields for blanks run closer in time to the samples presented in this study (the 2 blanks prior and following) range from 1.9 ug (PB060618, 6[th] of June, 2018) to 6.3 ug (PB071618, 7[th] of July, 2018). The measurements in this study were made during mid to late June 2018. Nonetheless, the range of the more proximal blanks (4.4 ug) still exceeds the maximum range in our five measurements (2.2 µg mentioned above). The difference in carbon yields could therefore be due to blank magnitude and variability. However, the measured $^{14}$C concentrations cannot be explained in the same manner, because the excess $^{14}$C is much higher than the $^{14}$C content of process blanks. We think this indicates that the observed differences in $^{14}$C concentration are not due to differences in blank magnitude and variability.

We have observed, multiple times, process blanks with a low carbon yield producing $^{14}$C concentrations that are higher than other blanks with higher carbon yields. We have included Table A1 at the end of the review response that includes information on the process blanks that have followed those included in Goehring et al. (2019) that used the same extraction method. An example of process blanks with a low carbon yield producing $^{14}$C concentrations that are higher than other blanks with higher carbon yields include PB101818 and PB110618 vs PB100118 in Table A1 below. Changes in blank carbon yield are evidently not always reflected in in situ $^{14}$C measurements.

*For these reasons, it would have been nice to have some indication on what the expected C-14 concentration in the samples being used for the experiment, is. Would it be possible to estimate based on Be-10 or other information what the expected C-14 in this rock material would be? This would lend more credibility to the results presented here.*

We agree that it would be helpful to estimate the $^{14}C$ concentration of the sample using the $^{10}Be$ concentration. Unfortunately, the sampling location, the island of Utsira, is a complicated locality to use $^{10}Be$ concentrations to estimate expected $^{14}C$ concentrations due to the problem of nuclide inheritance. We have measured the $^{10}Be$ concentration of the sample as part of another project ($8.10 \times 10^4$ atoms $g^{-1}$). If we use the $^{10}Be$ concentration, as well as the $^{14}C$ concentration of aliquots 1 and 2, to crudely calculate apparent exposure ages using the online calculator ([https://hess.ess.washington.edu/math/v3/v3_age_in.html](https://hess.ess.washington.edu/math/v3/v3_age_in.html)) we get a $^{10}Be$ age of ca. 17.5 ka and an in situ $^{14}C$ age of ca. 5.5 ka.

Inheritance of $^{10}Be$ has been observed in both bedrock and erratic samples on the island of Utsira (Svendsen et al., 2015 QSR; Briner et al., 2016 GRL). Svendsen et al. (2015) showed that the island deglaciated ca. 20 ka based on $^{10}Be$ dating of erratics and report a single $^{10}Be$ bedrock age of ca. 40 ka. Briner et al. (2016) then showed that the erratic $^{10}Be$ measurements of Svendsen et al. (2015) contain ca. 8000 atoms of $^{10}Be$, likely sourced from the deep production of $^{10}Be$ via muons during ice free periods and insufficient subglacial erosion.

Given the prevalence of $^{10}Be$ inheritance on Utsira, it is unsurprising that our $^{14}C$ age is younger than the $^{10}Be$ age. To complicate the matter further, the sample was collected from the top of a bedrock quarry which was recently exhumed, so there was likely material on top of the sampling location following the LGM that has since been removed, which would help explain the relatively young $^{14}C$ age.

The fact that the $^{14}C$ age is finite, rather than infinite, and that it is younger than the $^{10}Be$ age, is consistent with the previous studies.

*(3) Quartz isolation procedures*
*I would suggest a more careful formulation of the purpose of the froth floatation (referring to line 30) which never intended to replace density separation, and serves as a crude separation of feldspar minerals from quartz.*

We have removed the text on heavy liquid separation.

*Similarly, I was wondering whether there was any significance to the 'metal bowl' (line 80) used? Would a plastic bowl work?*

There is no significance in the use of the metal bowl and a plastic bowl would indeed work. We have altered page 3, line 96 to now say "A few drops of eucalyptus oil are added to the sample in a bowl (usually metal or plastic)…".

*Unfortunately, the manuscript does not provide information on how often the acid mixtures were changed during each of the steps. For example, in Table 1 during the 4 days on the shaker table (samples 1), was the acid mixture changed or the same HF/HNO3 was used for 4 days? This information would be useful if the authors wanted others to follow some of the recommendations provided.*

We agree that this information would be useful. We changed the acid mixture after each period that the aliquots spent on the shaker table and in the ultrasonic bath (so the acid mixture was changed approximately every 24 hours). Aliquots were rinsed prior to the addition of the new acid mixture. We have added this information to the text:

Page 6, lines 190 to 191:

"A new acid mixture was used with the samples following a set of rinses with ultrapure 18,2 MΩ water, such that each aliquot received a new acid mixture once every 24 hours."

*It should also be noted that some C-14 labs perform a concentrated HNO3 wash of the purified quartz at temperatures of 120 – 140 oC and this might well remove any residual laurylamine.*

As noted in the manuscript, we sonicate samples in 50% v/v $HNO_3$ prior to carbon extraction, but we do not heat the ultrasonic bath (see following response).

*At ETH, the HNO3 wash is followed by drying of the samples using an UV lamp. The authors note on line 220 that a HNO3 etch is performed but do not provide information on whether the samples are heated during the etch or not. Would this make a difference and did the authors look into that?*

We have added to the text that the sample is leached for 30 minutes in an unheated ultrasonic bath (page 9, lines 295 to 296).

*On a related note, the authors recommend the use of both shaker table and ultrasonic bath. Most labs will have one or the other and, again, it would have been useful – if the intention is to get people to adopt the recommendations presented here – to perform experiments for each of these (shaker table and ultrasonic bath) separately instead of changing the acid concentration between the two equipment.*

It is true that many labs may only have either a shaker table or an ultrasonic bath. Similarly, to the response directly following this one, it is likely that using a shaker table for four days would have the same effect as using an ultrasonic bath for the same amount of time. We have added the following to the text:

Page 9, lines 307 to 309:

"If a laboratory has only a shaker table or an ultrasonic bath, we would speculate that a minimum of four 24-hour periods in 5 % $HF/HNO_3$ would be sufficient to remove froth flotation-derived contamination.

*I am certain that if the ultrasonic bath samples would have been etched with 5% acid mixture the result would be identical.*

We agree that using 5% acid with the samples in the ultrasonic bath (for 4 x 24 hrs) would have probably produced identical, or thereabouts, results. The point of aliquots 4 and 5 was to use the absolute minimum duration of etching (and strength of acid/method of etching (shaker table or ultrasonic bath)) that we have found other laboratories using to isolate quartz, i.e. using 1% HF or $HF/HNO_3$ until samples appear to be pure quartz upon visual inspection. This is an important piece of context that we did not include, thus we have added more context for aliquots 4 and 5 to the text.

Page 6, lines 186 to 190:

"Aliquots 4 and 5 were not etched on the shaker table and both spent two days in the ultrasonic bath, after which they were visually pure, with the former etched in 1 % $HF/HNO_3$, and the latter etched in 1 % HF. Etching samples until quartz is visually pure is a common procedure used to isolate quartz for cosmogenic nuclide analysis. Aliquots 4 and 5 thus represent a feasible minimum duration of etching and were analysed to test if the short duration is sufficient to remove potential contamination."

Without the context of froth flotation and potential contamination, many laboratories do not necessarily have a standard procedure with a set number of days on a shaker table and/or in an ultrasonic bath but will etch samples until they appear to be solely composed of quartz. It would be a natural stopping point, especially if the samples were intended only for [14]C analysis and thus there wouldn't be any need for more intense etching to reduce major ion concentrations (e.g. Fe, Ti, and Al) for [26]Al and [10]Be measurements. We showed that, with our sample, it took only two days with 1% HF to produce visually pure quartz, which we would have previously been happy to use for in situ [14]C analysis. We think demonstrating that two days in an ultrasonic bath in 1% HF or $HF/HNO_3$ doesn't remove modern carbon introduced by froth flotation is an important result that other laboratories may find useful.

*It would also have been informative to present some ICP data on sample purity following the various steps.*

We agree that this would have been informative, but we do not think it is a completely necessary component.

*Further, I would also assume that the amount of laurylamine (or eucalyptol) that could get trapped in cracks in the quartz grains will also depend on the type of sample and history of cleaning prior to froth flotation – i.e., some quartz grains will be more damaged than others.*

We agree entirely and have added to the text on this issue.

Page 9, lines 320 to 323:

"The potential quantity of laurylamine or other contaminants able to reside in the microfractures of a particular sample will presumably vary with the lithology and geologic history of the sample, as well as the methods of sample preparation. The natural abundance of microfractures in a sample prior to sample collection will vary and microfractures may also be introduced during sample collection, crushing and milling."

*(4) Figures*
*The figures presented in this manuscript need a bit more work. Figure 1 is especially difficult to read and the use of colour or different symbols would help the reader. Also using arrows to guide the eye as to the direction in which points should be shifting, would help.*

We agree that colour would help with this figure. We have altered Figure 1 so that we now differentiate between the first, intermediate (second for the two samples measured three times), and final measurement using different coloured markers and think that this change makes the figure clearer.

*Also, should there be a table accompanying Figure 1? Or is this data published elsewhere?*

We did not publish the data for Figure 1 as we did not want it to be possible to identify the laboratories from which the anomalously high $^{14}C$ concentrations were sourced. The reason for this is that we would not want to potentially harm the reputation of any laboratory. Some of the data is also unpublished from a PhD student. We want the focus to be on identifying the cause of contamination and avoiding it.

*Figure 2: it is confusing, especially with the split into B and C.*

We agree that the figure was confusing in the initial submission and have removed panel C and altered panel B so that it now features a split y-axis, to show the finer details of all five aliquots on the same plot.

*(5) Table 2*
*Did the authors apply the same 6% uncertainty everywhere? The error on the number of atoms blank corrected is identical to the error on the effective blank. This simply cannot be.*

The 6 % uncertainty is applied to the $^{14}C$ concentrations. The reviewer has identified a mistake in our reported uncertainty for the total $^{14}C$ atoms blank corrected, and this has been corrected. The uncertainties are updated in the manuscript and the specific values now look like this:

| Aliquot Number | Total $^{14}$C atoms | ±1σ |
|:---:|:---:|:---:|
| | blank corrected | |
| 1 | 3.34E+05 | 9.439E+03 |
| 2 | 3.15E+05 | 9.548E+03 |
| 3 | 4.13E+05 | 1.029E+04 |
| 4 | 3.76E+05 | 9.845E+03 |
| 5 | 1.47E+06 | 2.230E+04 |

Thank you for noticing this error.

*(6) Technical comments*

*Line 150: 'to remove any adsorbed atmospheric CO2 and combust any carbon derived from handling and dust.' I would think that the removal of "dust" is entirely dependent on what is the dust made of and would be removed at 500 oC only if it is made of organic components. In most cases, however, dust is composed of inorganic particles.*

We agree with the reviewer and, upon revisiting the referenced Lifton et al. (2001) publication, we have removed reference to "dust" in the sentence under question.

*Line 160: 'Typical total analytical uncertainties are 1.5 to 2.5 % including the blank correction.' I recommend that authors remove this statement as it is incorrect as this depends on the activity of the sample and also the relative blank contribution. For example, in Table 2 in the current manuscript the blank correction is _16%. Wouldn't this have an effect on uncertainties?*

We have altered the text such that it now reads (Page 6, lines 206 to 207):

"Typical total analytical uncertainties are 1.5 to 2.5 %. Blank corrections, as a percentage of the total $^{14}$C atoms in each sample, range from 13.5 to 17.0 % (Table 2)."

*Line 175: 'with the unit yields, the 14C concentration of aliquots 1 and 2 are the same within uncertainties and are distinguishable from the unit yields of aliquots 3 to 5 when using the conservative 6 % uncertainty (Fig. 2). We observe elevated 14C concentrations for aliquots 3 to 5 relative to those of aliquots 1 and 2, with a particularly high 14C concentration for aliquot 5 (Fig. 2B). Figure 2 shows that the higher unit yields correspond with higher measured 14C concentrations.' If this statement is correct, shouldn't the unit yield for aliquot 5 be 2.5 ugC?*

We did not intend to suggest that there was a direct correlation between unit yield and $^{14}$C concentration, just that there was a general trend worth noting.

*Line 190 'Differing quartz isolation procedures used at other laboratories may therefore explain why quartz isolated from the same samples at Tulane and elsewhere produced vastly different 14C concentrations and unit yields (Sect. 1.2). ' Given the above points explained in detail I think this statement is only partially valid and would recommend a more careful explanation of the concentration differences.*

Given our responses to the above points, including the changes that we have added to the text in response to the review comments, we think that the statement being referred to is correct. We specifically use the word "may" to

ensure that we are not saying categorically that the differing quartz isolation procedures caused the differences in [14]C, but we think the evidence presented in the paper at least suggests it.

*Line 205-210; 'final measurement made for each sample is free from laurylamine contamination. For the samples presented in Fig. 1, the excess 14C concentrations range from 1.38 x 105 to 3.23 x 105 at g-1. The associated residual carbon ranges from 2.32 to 5.42 _g g-1, and the residual laurylamine ranges from 2.98 to 6.96 _g g-1, both per gram of quartz. We speculate that the latter residual carbon and laurylamine estimates, an order of magnitude greater than those presented in this study.' 7 ug of*
*laurylamine /gram of quartz sounds like a large number. I was wondering whether the authors have considered other potential sources of the excess C. Perhaps it could be related to fluid inclusions or other minerals present in the sample that are only removed following additional HF leaching?*

We agree that 7 ug of laurylamine/gram of quartz is very high and were initially surprised by the value. We did not originally report the scale of the change in the carbon yields in the initial submission, but the initial anomalous results were associated with high yields on the order of many 10s of ug of carbon, with large reductions in carbon yield upon repeat measurement (with one exception). We have added a new figure (now Fig. 2) which shows the initial and final unit yields associated with the same measurements presented in Fig.1.

Fig. 2:

[Figure]

We have added the following caption for the new Fig. 2 (Page 12, lines 426 to 428):

"Figure 2: Initial and final unit yields associated with the same measurements presented in Fig.1. The initial unit yield measurements for each sample were made using quartz isolated at external laboratories, whilst the final unit yield measurements were made using quartz isolated at Tulane using our standard procedure. Error bars are smaller than the data points."

We do not include the data presented in Fig. 2 for the same reasons we describe in an earlier response regarding Fig. 1.

We refer to Fig. 2 here (Page 4, lines 140 to 141):

"With the exception of one sample, carbon yields were reduced (Fig. 2), and for all samples the resulting $^{14}C$ concentrations were both lower and geologically plausible (Fig. 1)."

The scale of change in both $^{14}C$ concentration and carbon yield was much greater than the scale of change we have found in the measurements made for this study.

We are confident that the measurements in Fig. 1 were made using pure quartz and that the elevated carbon concentrations were not sourced from other minerals present in the sample. The samples in Figs. 1 and 2 were run for ICP-MS to test their suitability for $^{10}Be$ dating, thus were assumed to be sufficiently pure quartz for $^{14}C$ analysis.

We have added that to the text here (Page 5, lines 150 to 154):

"Measurements presented in Figs. 1 and 2 were made using quartz which was not only visually pure but had initially been isolated for $^{10}Be$ measurements. The samples had previously been sent for ICP-MS analysis to test their suitability for $^{10}Be$ analysis, confirming that they were comprised of sufficiently pure quartz and thus were ready for $^{14}C$ analysis as well. We are therefore confident that the elevated $^{14}C$ concentrations were not sourced from other minerals that persisted through quartz isolation."

And here (Page 8, lines 281 to 283):

"As noted in Sect. 1.2, we are confident that the elevated $^{14}C$ concentrations were not sourced from other minerals that persisted through quartz isolation because the quartz separates were previously analysed by ICP-MS to confirm their suitability for $^{10}Be$ analysis."

Presumably, fluid inclusions would not contain $^{14}C$ due to the short half-life of the isotope relative to the crystallization and Caledonian age deformation. If the reviewer is referring to the potential production of $^{14}C$ on $^{14}N$ contained in fluid inclusions, a very large abundance of $^{14}N$ would be required in said fluid inclusions. Fluid inclusions would likely influence total carbon yields though. In fact, this is something we have noticed through measurements made in our laboratory. Samples of quartz sourced from metamorphic rocks often yield noticeably larger carbon yields than samples from igneous or sedimentary rocks.

We have added the following to the text here (Page 8, lines 283 to 287):

"Though fluid inclusions may contribute to elevated carbon yields, they would presumably be devoid of $^{14}C$ and thus could not explain the anomalous $^{14}C$ concentrations. Production of $^{14}C$ on $^{14}N$ in fluid inclusions through thermal neutron capture is possible, however, the presumably low abundance of $^{14}N$ means that this production mechanism is unlikely to contribute significantly to $^{14}C$ concentrations when compared to the spallation component (Lal and Jull, 1998)."
* * *
**Reviewer 2**

*1. In lines 150-152, it is stated that the samples are diluted with (presumably dead) CO2. In table 2, this appears to be corrected for the dilution, but the values given for 14C/13C appears to be $8.47 \times 10^{-12}$ to $3.64 \times 10^{-11}$ which must be incorrect. Modern carbon is about $10^{-10}$ 14C/13C. The value stated in the paper for the laurylamine is 1.03 times modern (i.e. about $1.2 \times 10^{-12}$ 14C/12C. for 14C/13C this should be around $10^{\sim 10}$), so even if the sample was 100% the contaminant this would still be wrong. I assume this is some arithmetic error but it needs to be corrected.*

The "corrected" in the column heading "14C/13C corrected" refers to the 14C/13C ratio corrected for the contribution by graphitization as discussed in Goehring et al. (2019) and elsewhere (e.g., Slota, 1987). We apologise for the misleading column heading, which we have updated (see response directly below).

*2. In table 2, an explanation of the various columns would be helpful.*

We agree that the table column headings were unclear. We have altered some of the headings and have added to the table caption such that it now reads (Page 12, lines 443 to 449):

"Table 2: In situ $^{14}$C analytical data. Aliquot number is described in the text. See Table 1 for the different quartz isolation procedures used for each aliquot. TUCNL is a unique sample identifier for each sample analysed at the Tulane University Cosmogenic Nuclide Laboratory. C yield is the carbon yield prior to dilution. Unit yield is the carbon yield divided by the quartz mass. Total $^{14}$C blank corrected is the total number of $^{14}$C atoms corrected using the effective blank. Effective blank is representative of the blank during the running of the samples presented. See Sect. 2 for rationale behind the use of the 6 % uncertainty for the $^{14}$C concentrations. We also include 1σ uncertainty for the $^{14}$C concentrations for completeness. The mass of residual carbon and laurylamine for aliquots 3 to 5 are calculated using the $^{14}$C/$^{12}$C ratio of laurylamine as measured (see Sect. 4)."

*3. In table 2, a value of d13C ca. -5 per mil is given. I assume this is of the diluted (not undiluted) gas?*

Yes, your assumption is correct.

*4. The authors also note that the procedure involves adding the laurylamine to acetic acid. Yet, the acetic acid can be either from biogenic or nonbiogenic sources. Was this tested for 14C? 5. The authors might wish to review the chemistry of this process and the different phases that can form, for example there is a paper by S. Karlsson et al. (2001) Phase Behavior and Characterization of the System Acetic Acid-Dodecylamine- Water, Langmuir 17, 3573.*

As per our response to reviewer #1, this is a great point brought up, that we did not test acetic acid for $^{14}$C. Thank you for bringing our attention to the study by Karlsson et al. (2001), this was an informative paper and we have consequently added to the text. Our rationale for not testing acetic acid or eucalyptol for $^{14}$C, as well as reference to the behavior of acetic acid and laurylamine, has been added to the text here:

Page 4, lines 154 to 165:

"We suspected that the froth flotation procedure was a potential source of $^{14}$C contamination because it involves the introduction of carbon to sample material through the use of three aforementioned compounds. We focused on the long-chain compound laurylamine because eucalyptol is volatile at room temperature and is thus unlikely to persist through sample etching. Acetic acid is predominantly sourced from methanol which is, in turn, largely derived from $^{14}$C dead natural gas, though it can be produced using modern material and therefore may have the potential to contaminate samples with $^{14}$C. However, regardless of the source, acetic acid is a simple compound that would be relatively easy to break down during etching when compared to laurylamine. There is a complicating factor, in that acetic acid and laurylamine can form complex molecules that behave as a singular species (Karlsson et al., 2001), which may increase the potential for acetic acid to remain on sample material after froth flotation and contribute to potential $^{14}$C contamination. Again, though the predominantly $^{14}$C dead source material minimises potential acetic acid influences. Nonetheless, we focused on laurylamine but acknowledge that it may not be the sole contributor to residual $^{14}$C following froth flotation."

As per our response to comments from reviewer #1, we have altered the text to take a more general approach whenever referring to laurylamine contamination. This is because we did not test acetic acid of eucalyptol for $^{14}$C, a great point brought up by both reviewers. We now state that laurylamine has the potential to contaminate samples with modern carbon, but we do not know if it is necessarily that specific compound (or another factor) that is contaminating samples. We state wherever needed that froth flotation in general is contaminating samples, not necessarily only laurylamine. All examples of this change in the text are given in our response to reviewer #1, but the most important change is probably this one on page 7, lines 235 to 238:

"We did not measure the Fm of acetic acid or eucalyptol due to the rationale described above (Sect. 1.2) and thus we cannot rule out their potential to contaminate samples with $^{14}$C. However, the modern carbon source of laurylamine

confirms that the froth flotation procedure, regardless of the contributing compound, introduces $^{14}$C to sample material."

**Additional Changes**

In addition to the changes made to the manuscript in response to reviewer comments, we have also added two new figures (4 and 5). The new figures present SEM images of quartz grains from each aliquot, as well as an unetched sample, to provide additional information regarding the hypothesis that contamination may be residing in microfractures.

Fig. 4:

[Figure]

Fig. 5:

[Figure]

We have added the following figure captions for Figs. 4 and 5 (Page 12, lines 434 to 437):

"Figure 4: SEM images of quartz grains of an unetched sample and aliquots 1 and 2. Red boxes on the left show the location of the adjacent image to the right. The unetched sample is sourced from the same whole rock sample as the five aliquots and was crushed, milled, sieved, rinsed and magnetically separated. Note the conchoidal fracture in B."

Figure 5: SEM images of quartz grains of aliquots 3 to 5."

We have also added the following text to pages 9 to 10, lines 323 to 353:

"Figures 4 and 5 show evidence of microfractures on the surface of quartz grains from all aliquots. In addition, Fig. 4 shows a quartz grain from an unetched aliquot that was sourced from the same whole rock sample as the five aliquots. Anecdotally, whilst using the SEM we observed microfractures that seemed to be opened up to a greater extent in aliquots 1 and 2, which received the longest duration of etching, compared to aliquots 3 to 5. Note the high surface roughness of the unetched sample (Fig. 4a and b) and the relative smoothness of the grains in all aliquots (Figs. 4 and 5), a result of the partial dissolution by HF of quartz grains which will have presumably removed some microfractures entirely. We observe that further etching, both in our initial measurements (Sect. 1.2) and when comparing aliquots 2 and 3 with aliquots 4 and 5, lowers carbon yields and $^{14}$C concentrations. The longer duration in acid may indicate that the HF is opening up microfractures and allowing contamination to be more thoroughly removed, highlighting the importance of HF in the removal of contamination, though this would be difficult to test, and an extensive systematic study would be required to make conclusions with any statistical significance. Whilst the presence of microfractures does not confirm our hypothesis, Figs. 3 and 4 do show that there are abundant microfractures and surface features for contaminants to potentially reside in following froth flotation."

We have also added to the acknowledgements section and have fixed a number of minor spelling and grammatical mistakes.

**Table A1:**

| Sample ID | TUCNL# | C yield (µg) | ±1σ (µg) | Diluted Gas Mass (µg) | ±1σ (µg) | $^{14}C/^{13}C$ | ±1σ | $\delta^{13}C$ (‰) | ±1σ | $^{14}C/C$ total | ±1σ | $^{14}C$ atoms (at) | ±1σ |
|---|---|---|---|---|---|---|---|---|---|---|---|---|---|
| PB051418 | TUCNL-295 | 3.1 | 0.04 | 95.6 | 1.2243 | 1.50E-12 | 3.61E-14 | -4.21 | 0.5 | 1.64E-14 | 3.96E-16 | 7.87E+04 | 2.15E+03 |
| PB060618 | TUCNL-304 | 1.9 | 0.02 | 90.9 | 1.1641 | 8.40E-13 | 2.38E-14 | -3.94 | 0.5 | 9.22E-15 | 2.61E-16 | 4.20E+04 | 1.31E+03 |
| PB070818 | TUCNL-313 | 5.1 | 0.07 | 88.2 | 1.1295 | 2.47E-12 | 4.48E-14 | -4.13 | 0.5 | 2.71E-14 | 4.92E-16 | 1.20E+05 | 2.66E+03 |
| PB071618 | TUCNL-318 | 6.3 | 0.08 | 120.1 | 1.538 | 2.42E-12 | 3.43E-14 | -3.29 | 0.5 | 2.65E-14 | 3.77E-16 | 1.60E+05 | 3.06E+03 |
| PB080818 | TUCNL-327 | 3.1 | 0.04 | 116.4 | 1.4906 | 9.45E-13 | 2.22E-14 | -2.98 | 0.5 | 1.04E-14 | 2.44E-16 | 6.06E+04 | 1.62E+03 |
| PB082318 | TUCNL-336 | 7.2 | 0.09 | 105.3 | 1.3485 | 2.67E-12 | 3.75E-14 | -3.4 | 0.5 | 2.94E-14 | 4.12E-16 | 1.55E+05 | 2.95E+03 |
| PB100118 | TUCNL-345 | 3.7 | 0.05 | 103.2 | 1.3216 | 1.37E-12 | 2.92E-14 | -3.92 | 0.5 | 1.50E-14 | 3.21E-16 | 7.78E+04 | 1.94E+03 |
| PB101818 | TUCNL-354 | 4.5 | 0.06 | 88.8 | 1.1372 | 2.16E-12 | 5.13E-14 | -3.69 | 0.5 | 2.37E-14 | 5.63E-16 | 1.06E+05 | 2.85E+03 |
| PB110618 | TUCNL-363 | 4.1 | 0.05 | 277.3 | 3.5512 | 7.06E-13 | 2.68E-14 | -1.94 | 0.5 | 7.76E-15 | 2.95E-16 | 1.08E+05 | 4.33E+03 |